# Mentha and Oregano Soil Amendment Induces Enhancement of Tomato Tolerance against Soilborne Diseases, Yield and Quality

**Kalliopi Kadoglidou [1],\*, Paschalina Chatzopoulou [1], Eleni Maloupa [1], Argyrios Kalaitzidis [1], Sopio Ghoghoberidze [2]** and **Dimitrios Katsantonis [1]**

[1]  Institute of Plant Breeding and Genetic Resources, Hellenic Agricultural Organization-Demeter, 57001 Thermi Thessaloniki, Greece; chatzopoulou@ipgrb.gr (P.C.); director@ipgrb.gr (E.M.); kalaitzidis@cerealinstitute.gr (A.K.); dikatsa@cerealinstitute.gr (D.K.)

[2]  Institute of Phytopathology and Biodiversity, Batumi Shota Rustaveli State University, Batumi 6010, Georgia; sopio.ghoghoberidze@bsu.edu.ge

\*  Correspondence: kkadogli@agro.auth.gr; Tel.: +30-231-0421-199; Fax: +30-231-0471-544

**Abstract:** Tomato is considered among the most important horticulture crops in both temperate and tropical regions, and two of the major biotic stresses include *Fusarium oxysporun* f.sp. *lycopersici* (*Fol*) and *Verticillium dahliae* (*Vd*). The effect of soil incorporated spearmint and oregano dried plant material on physiological, yield and quality parameters of tomato, along with their efficacy against soilborne fungal diseases, was studied in pot experiments conducted in a greenhouse environment. Tomato plants grown in soil amended with spearmint or oregano showed better agronomical characteristics (taller plants with thicker stems) and improved physiological ones (higher chlorophyll content index and photosynthetic rate). Yield was increased and the qualitative features of tomato fruits were enhanced. In addition, plants grown in soil amended with spearmint or oregano and inoculated with *Fol* or *Vd* had no visual disease symptoms 50 days from the inoculated tomato transplantation, except from plants grown in soil amended with oregano and inoculated with *Fol*, which showed symptoms of chlorosis and leaves loss. These enhancements on physiological parameters and on disease suppression resulted in increased fruit yields of plants–grown in soil amended with spearmint and oregano and inoculated with *Fol* or *Vd*–by 77%–95% compared with free-disease controls. GC-MS analysis of volatiles derived from soils amended of either spearmint or oregano indicated that several constituents remained in the soil environment long after incorporation of plant material, although, at lower concentrations and considerable modified. The current study reveals that direct incorporation of spearmint or oregano plant material into the soil could improve tomato tolerance against soilborne fungi, soil fertility and consequently increase yield and product quality.

**Keywords:** antifungal tool; plant protection; renewable inputs; soil amendment; sustainable disease management

## 1. Introduction

Numerous studies in the last decades have been dedicated to the exploitation of essential oils and their constituents in modern sustainable agriculture systems as possible alternatives to synthetic pesticides. Essential oils and other secondary metabolites were shown to have antifungal [1,2], antibacterial [3] insecticidal, and herbicidal activity [4–6] in vitro. There is an increasing interest on the activity of allelopathic compounds isolated from plants and their use as alternatives to synthetic fungicides [7]. Such a shift in the research field came from the development of resistance of many pathogens to synthetic fungicides [8–10] and their negative effect on beneficial organisms [11].

Moreover, the use of a significant number of synthetic fungicides has been restricted or banned in many countries [8,12]. Thus, the need to find safer alternative methods of plant protection is imperative.

Fungicides based on secondary metabolites of aromatic plants could be used as alternatives for disease control, and be particularly useful in organic farming systems. Many researchers have studied the use of microencapsulated essential oils as bioagrochemicals [13–16]. Another possible way to exploit the allelopathic properties of aromatic plants is their utilization as compost, as green manure or as companion plants. Although such methods have received comparatively less attention, several studies have pointed to their potential benefits. The incorporation of aromatic plants into the soil—in the form of compost or as green manure—was proved by many researchers to enhance: (a) soil fertility [17–19], and (b) provide an allelopathic action against weeds [6,20,21]; whereas De Carvalho et al. [22] reported an increase in tomato yield in an intercropping system with basil, rue, fennel or mint.

Tomato is among the ten most important crops of South Eastern Europe [23]. Vascular wilts, caused by *Vd* and *Fol*, are considered among its most important diseases of tomato. Disease management is difficult due to the endophytic growth of the pathogens and their persistence in soil. Preventive measures such as soil solarization, use of steam, crop rotation and application of biocontrol agents, such as *Trichoderma asperellum* and *T. gamsii*, provide only moderate control [24]. The use of resistant cultivars has been the most effective control method [25], but the occurrence and evolution of new pathogenic races is a continuous problem. Fungicidal or fungistatic effects have been observed in vitro against different species of *Fusarium* and *Verticillium* [1,2,12], whereas La Torre et al. [26] reported that rosemary, clove, and thyme oils were able to reduce *Fusarium* wilt of tomato in a greenhouse pot experiment. In a previous in vitro study [2] essential oils from spearmint and oregano—two species abundantly present in Greece [27,28]—as well as their main components carvone and carvacrol, inhibited both *Fol* and *Vol*. Additionally, the incorporation of spearmint compost improved the properties of soil along with the physiology of tomato seedlings [17], whereas that of raw spearmint enhanced the production of tomato seedling in seedbeds [19]. Nevertheless, the application of raw material of spearmint and oregano in tomato cultivation in greenhouse or field conditions has not been investigated yet.

Thus, the aim of this study was to evaluate the potential of spearmint and oregano used as: (1) antifungal tool against *Vd* and *Fol*, and (2) as soil amendment, that improves yield and quality indices in tomato cultivation.

## 2. Materials and Methods

Greenhouse experiments were carried out in pots, at the establishments of Institute of Plant Breeding and Genetic Resources (IPGRB) of the Hellenic Agricultural Organization-Demeter (Demeter) (Thessaloniki, North Greece, 40°32′11.69″ N and 22°59′58.08″ E). The greenhouse size was $10 \times 20$ m and it was covered with 40% shading net.

### 2.1. Production of Plant Material

Two kinds of aromatic plants were used as soil amendments in the present study, spearmint *(Mentha spicata* L.) and greek oregano (*Origanum vulgare* L. subsp. *hirtum)*. For both species, the whole aboveground biomass (shoots and leaves), were harvested from an experimental field of IPGRB. After collection all plant material was cut into small pieces (approximately $1 \times 1$ cm), air-dried in the dark until the moisture content reached 5%–7%, and stored for two weeks in dry conditions at 12 °C in the dark until use [19].

### 2.2. Soil Amendment and Decomposition of Plant Material

2.2.1. Soil Properties and C/N Ratio

The soil used in the present study was collected from the top layer (0–30 cm) of a fallow organic field of IPGRB, which was left for an eight-year period. Soil samples were analyzed in the Soil Science Institute of Demeter, consisted of 14% clay, 26% silt, 60% sand, 1.64% organic matter, 6.3% $CaCO_3$

and pH 8.1. The collected soil was sieved through a 3 mm sieve, weighed and mixed with plant material of either spearmint or oregano, at rate of 4% (w:w, plant material:soil). The soil-aromatic plant mixture was distributed in 6 L pots and placed immediately in a glasshouse, in order to accelerate the decomposition process to reduce the C/N ratio, at 18 to 28 °C. This practice circumvents the arduous composting process and therefore, it can be more cost and time effective compared to the standard composting techniques. Pots containing non-amended soil were used as controls. For each one of the three types of soil treatment 36 pots were used. All pots were irrigated weekly. The pots were kept in glasshouse until decomposition of the plant material used as amendment and then used to grow tomato plants firstly produced as described in Section 2.3.

The C/N ratio of the soil was used as indicator of the decomposition process. A quantity of 10 g soil samples were taken at 0, 15, 30, 60 and 90 days after the incorporation of aromatic plants into the soil (DAI). Organic carbon (and subsequently organic matter) content was determined by the wet oxidation method [29] and total nitrogen content was determined by the macroKjeldahl method [30].

### 2.2.2. Volatiles Constituents in Amended Soil

The presence of volatile constituents from spearmint and oregano was also evaluated at 15, 30, 60 and 90 DAI. Thus, another two soil samples of 2 kg each were collected per soil treatment. These samples derived by surplus pots maintained within the main experimental plot, treated as all of the rest ones and to be used for the GS analysis. Soil samples were subjected to 3 h hydrodistillation, using a Clevenger-type distillation apparatus. Additionally, 100 g of the initial dry biomass of spearmint and oregano were collected at 0 DAI and subjected to the same process described above. The yield of essential oils was expressed in μL per 100 g of soil (dry weight) and compared to the initial plant material. The essential oils were collected and dried over anhydrous sodium sulphate prior to the analysis of their chemical composition. The analysis was performed by GC on a fused silica DB-5 column, using a Shimadzu GC-17A gas chromatograph interfaced with a Shimadzu QP-5050A mass spectrometer, and supported by the GC/MS Solution, Ver1.21 software. The chromatographic conditions were: Injection temperature: 260 °C; interface heating: 300 °C; ion source heating: 200 °C; EI mode: 70 eV; scan range: 41–450 amu, and scan time 0.50 s. The temperature programs were: (i) 55–120 °C (3 °C/ min), 120–200 °C (4 °C/min), 200–220 °C (6 °C/min) and 220 °C for 5 min; and (ii) 60–240 °C at 3 °C/min; Carrier gas: He, 54.8 kPa; split ratio 1:30. The percentage composition of the oils was computed after 3 GC runs of each sample from the peak areas without correction factors. The identification of the constituents was based on comparison of their retention indices (RI) relative to n-alkanes, with corresponding literature data, and by matching their spectra with those of the MS libraries (NIST 98, Wiley) [31].

### 2.3. Production of Tomato Seedlings and Transplanting to the Amended Soil

The tomato (*Lycopersicon esculentum* L.) variety "Early Pack 7" was used in all the experiments after preliminary tests due to the high level of susceptibility, similarly to Papadaki et al. [32]. Seedlings were produced in two seed trays of 144 cells each (volume 343 cm$^3$ /cell), filled with peat. In each cell two tomato seeds were sown, whereas thinning was carried out 10 days after sowing to maintain a ratio of one seedling per cell. The trays were placed in a growth chamber under a 16 h photoperiod at 22–24 °C, 60% relative humidity (RH) and light intensity of 450 μmol m$^{-2}$ s$^{-1}$ during day and 17–19 °C and 80% RH during the night, whereas irrigation was performed every other day. After three weeks and when seedlings reached the third to fourth leaf stage, they were used for the greenhouse experiment.

### 2.4. The Production of Inoculum of Verticillium/Fusarium

At this stage, preliminary experiments were carried out in order to confirm the pathogenicity of two fungi and to determine the most efficient inoculation method. The strains K.E. F421 of *Fol* race 2 and 2681 of *Vd* were obtained from the collection of the Benaki Phytopathological Institute, Athens, Greece, whereas initially the fungi were started as single spore cultures. Inoculum was produced in

500 mL Erlenmeyer flasks containing 250 mL of liquid medium: Potato Dextrose Broth (PDB) for *Fol* and Czapek Dox Broth (CDB) for *Vd*. Each flask was inoculated with 10 mycelial plugs of 10 mm diameter and incubated in the dark on a rotary shaker (80 rpm) for 6–8 days at 22 °C. The mycelial plugs were excised from 8–10 days old cultures incubated in the dark at 22 °C, grown on PDA for *Fol* and Czapek Dox Agar for *Vd*. After the incubation period the liquid cultures were agitated using a sterile blender in order to break up agglomerates of possible existing filamentous hyphae. Then the concentration of the conidial suspensions was adjusted to $10^6$ conidia mL$^{-1}$ using a haemocytometer.

## 2.5. Seedlings Inoculation and Cultivation

The tomato seedlings described in Section 2.3 were removed from the cells and soil was gently washed out using distilled water. Then the roots were dipped in spore suspensions of either *Fol* or *Vd* and kept for 10 min on a rotary shaker at 60 rpm [33]. Afterwards, the inoculated seedlings were transplanted into the pots described in Section 2.2 using one plant per pot. Non-inoculated seedlings used as controls and were transplanted in the pots. In order to facilitate the fungus development in the pots, they were maintained for seven days in a growth room at 22–24 °C under a 14 h photoperiod. Then they were placed in a structure covered with insect-proof netting, which will be referred to as "net-greenhouse" in the rest of this paper (Figure 1). The 108 pots of each experiment were arranged in rows of 0.5m distance pot by pot and 1m row by row. The temperature and relative humidity were recorded continuously in the net-greenhouse during the experiment and ranged from 18 to 35 °C and from 38% to 82%, respectively. The pots were drip irrigated with 200 mL/pot twice a day. The pots were hand-weeded and all the other culture practices were in accordance to the principles of organic farming. No pesticide treatment was conducted during experiment.

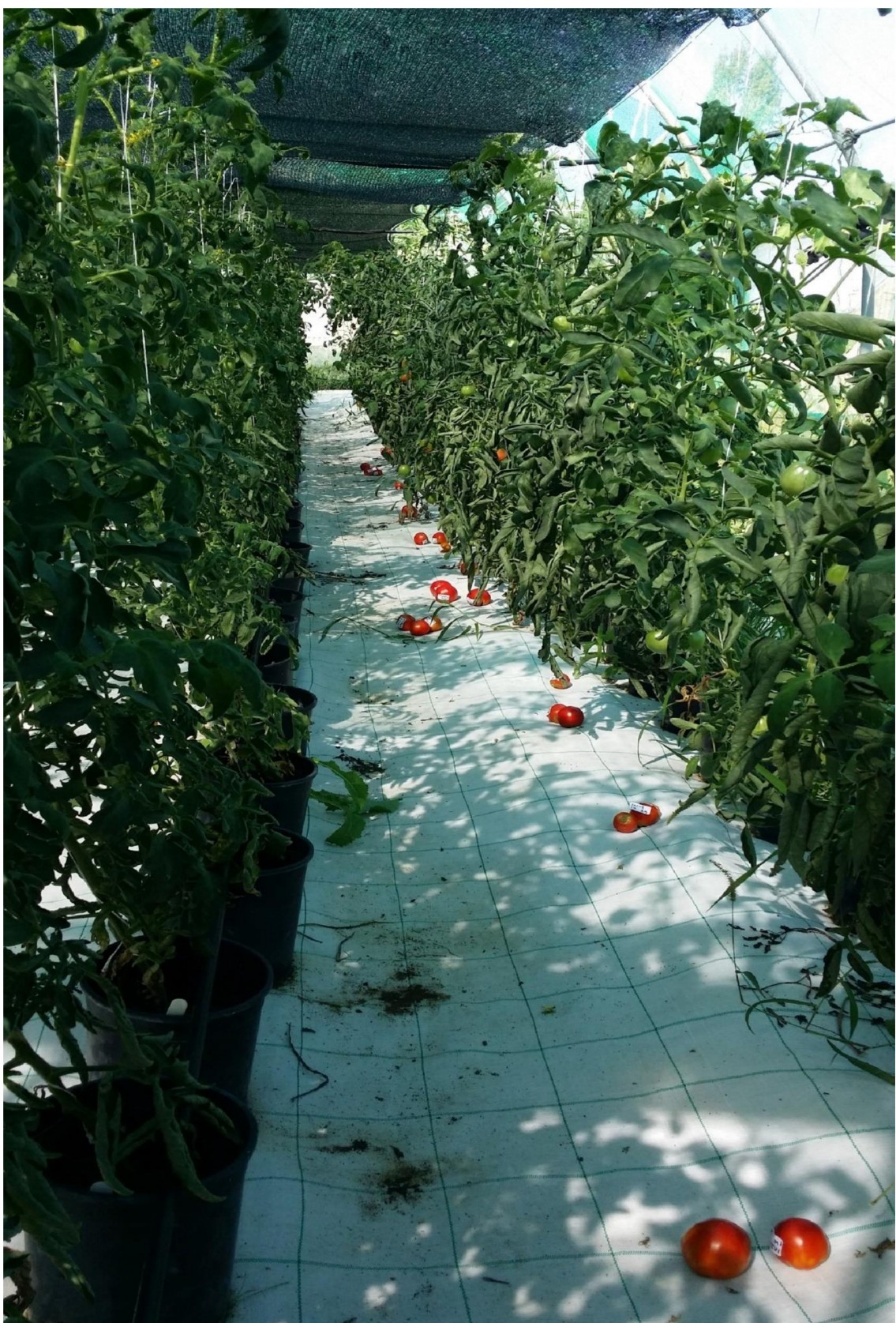

**Figure 1.** A view of the experiment in the net-greenhouse.

### 2.6. Disease Assessment

Disease was assessed 15, 30 and 50 days after seedlings transplanting (DAT), using disease severity assessment scale 1-6, where 1 = no symptoms, 2 = epinasty, 3 = chlorosis / losses of primary leaves, 4 = chlorosis / loss of leaf area < 50%, 5 = chlorosis / loss of leaf area > 50%, 6 = dead plants [34]. Based on this scale, an average degree of infection (ADI) was calculated for each plant and treatment, using the formula:

$$ADI = \frac{\text{total rating (sum of all individual assesments)}}{\text{number of inoculated plants per treatment}}$$

In order to better understand the effect of treatments on the diseases development the quantitative summary of disease intensity over time was determined as the area under diseases progress curves (AUDPC) that was estimated according to Shaner and Finnay [35] using the following formula:

$$\sum_{i=1}^{n} [(Xi + 1 + Xi) / 2][ti + 1 - ti]$$

$X_i$ = disease severity at the $i^{th}$ day, $t_i$ = the time in days of the $i^{th}$ observation, $n$ = the total number of observations.

At the end of the experimentation, the presence of a brownish discoloration around the vascular bundles was checked, by examining a cross section above the ground level at the plant stems.

### 2.7. Plant Growth and Physiology Assessments

Plant measurements were carried out at 12, 24, 36, 50 and 65 DAT including: (1) stem thickness, measuring with an electronic calliper at the point just beneath the third expandable leaf, (2) chlorophyll content index (CCI), measuring with a portable Chlorophyll Content Meter (CCM-200, Opti-Sciences, Tyngsboro, MA, USA). Moreover, net photosynthetic rate (Anet, $\mu$mol m$^{-2}$s$^{-1}$) was measured at 15, 30, 45 and 60 DAT, using the portable photosynthesis system LCi-SD (ADC Bioscientific Ltd., Hoddesdon, UK). Twelve measurements were taken per treatment and per experiment [one measurement by each plant (pot)]. The procedure was performed at an ambient $CO_2$ concentration of 400 ± 30 $\mu$mol m$^{-2}$s$^{-1}$, a temperature of 22/26 °C and a photon flux density of 800 ± 100 $\mu$mol m$^{-2}$s$^{-1}$. All the physiological parameters were measured on the upper third of the second fully developed leaf, counting from the plant apex, with the same orientation to the sun.

### 2.8. Fruit Production and Quality Evaluation

The harvesting of tomato fruits started on the 10th of August and was completed on the 25th September. A total of six harvests were carried out during the course of each experiment and fruit yield was weighted for each plant and fruit pH was determined. The content of soluble solids (°brix) was measured using a digital handheld refractometer (DR201-95 Krüss Optronic, Hamburg, Germany). Finally, dry matter (g/100 g fresh fruit) was assessed by oven drying at 72 °C for 48 h.

### 2.9. Experimental Design and Statistical Analysis

A randomized complete block design was used with 12 replicates (pots) for each one of the nine combined treatments. There were 3 treatments with incorporated material into the soil: (i) 4% (w:w) spearmint, (ii) 4% oregano, and (iii) 0% (untreated control), and 3 treatments inoculated with the fungi (i) *Fol*, (ii) *V.d*, and (iii) non inoculated plants. The entire experiment was repeated twice.

A combined analysis of variance was performed with combined data from the two greenhouse experiments (ET, experiments repeated in time) to test the 3 × 3 (three treatments with incorporated material × three fungal inoculation treatments) factorial arrangement, DAT being a sub-plot factor (repeated measures). Since the combined over the two field experiments analysis of variance had indicated no differences between them, the presented treatment means were averaged over the two

experiments. In all statistical analysis throughout the study, the LSD procedures were used to detect and separate mean treatments differences at level of $p < 0.05$ (Bonferroni test). The combined analysis of data was justified following the Bartlett's test for homogeneity of variances, which indicated that data were not heterogeneous. All statistical analyses were performed using STATISTICA software (ver. 7.06, Statsoft Inc). The different types of treatments and their abbreviations are presented in Table 1.

**Table 1.** Treatments applied and their abbreviations.

| Treatment Abbreviations | Treatments |
| --- | --- |
| C | Control non-amended soil + non inoculated seedlings |
| CF | Control non-amended soil + seedling inoculation with *Fol* |
| CV | Control non-amended soil + seedling inoculation with *Vd* |
| M | Soil + 4% (w/w) *M. spicata* plant material+non inoculated seedlings |
| MF | Soil + 4% (w/w) *M. spicata* plant material+seedling inoculation with *Fol* |
| MV | Soil + 4% (w/w) *M.spicata* plant material+seedling inoculation with *Vd* |
| OR | Soil + 4% (w/w) *O. vulgare* subsp. *hirtum* plant material+non inoculated seedlings |
| ORF | Soil + 4% (w/w) *O. vulgare* subsp. *hirtum* plant material+seedling inoculation with *Fol* |
| ORV | Soil + 4% (w/w) *O. vulgare* subsp. *hirtum* plant material+seedling inoculation with *Vd* |

## 3. Results

### 3.1. Analysis of Soil Characteristics after Amendment with Aromatic Plants

#### 3.1.1. C/N ratio

At the beginning of the experiment (0 DAI) C/N ratio in soil where 4% spearmint or oregano was incorporated was above 27, the highest reduction occurred in the first 15 DAI, and then downtrend was recorded with negligible fluctuations (Figure 2). For both plant species used as soil amendment the completion of the decomposition process was estimated to have occurred 60 days after soil amendment, when C/N ratio was stabilized at values lower than 20. At this time, the soil was used for transplanting tomato seedlings. There was no further significant evolution of C/N ($p > 0.05$) until the last assessment 30 days later (Figure 2). Similarly, from 0 to 90 DAI, the organic matter content decreased by 5.75% to 3.47% and by 3.89% to 2.64% for spearmint and oregano, respectively (data not shown).

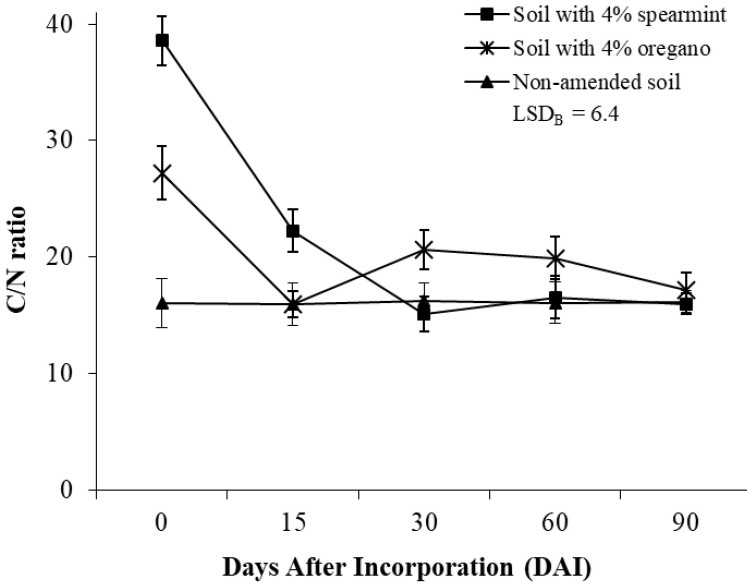

**Figure 2.** Changes in the C/N ratio in soil samples taken at different sampling times after the incorporation of 4% (w:w) spearmint or oregano plant material and in non-amended soil. Full description of the treatments examined is found in Table 1.

In our study, the C/N ratio was stabilized very soon (approximately in 60 days) at values below 20, revealing that the maturing of compost was achieved.

### 3.1.2. Volatiles in Soil Samples

Incorporation of 4% spearmint and oregano in the soil, caused significant changes in the quantitative (Table 2) and qualitative composition (Tables 3 and 4) of the volatiles in the distilled material compared to those obtained by the initial plant material of spearmint and oregano.

**Table 2.** Evolution over time of the total amounts of essential oils extracted from samples of soil amended with 4% (w/w) of spearmint or oregano tissue and kept in a glasshouse in order to accelerate the decomposition process.

| Days After Incorporation (DAI) | Total Yield of Essential Oil (µL/100 g of Soil) | |
|---|---|---|
| | **Spearmint** | **Oregano** |
| 0 | 129.10 | 270.30 |
| 15 | 24.00 | 35.50 |
| 30 | 4.16 | 14.50 |
| 60 | 2.37 | 6.75 |
| 90 | 0.50 | 1.80 |

**Table 3.** Qualitative and quantitative analysis of volatiles present (i) in spearmint plant material, and (ii) in soil samples taken at different times after amendment with 4% (w/w) spearmint tissue. Data averaged over two independent experiments.

| Identified Compounds | *RRI*[a] | *LRI*[b] | Concentration (%)[c] | | | | |
|---|---|---|---|---|---|---|---|
| | | | (i) Spearmint | (ii) Soil Amended with Spearmint | | | |
| | | | | Days After Soil Amendment | | | |
| | | | | 15 | 30 | 60 | 90 |
| α-pinene | 937 | 937 | 0.96 | 1.92 | 2.68 | 1.48 | |
| sabinene | 976 | 976 | 0.64 | 1.30 | 0.82 | 0.38 | |
| β-pinene | 979 | 980 | 1.23 | 2.47 | 3.26 | 1.83 | |
| β-myrcene | 993 | 991 | 0.81 | 1.28 | 0.49 | | |
| α-terpinene | 1018 | 1018 | 0.48 | | | | |
| limonene/β-phellendrene | 1030 | 1031 | 10.80 | 18.49 | 9.61 | 3.95 | |
| 1,8-cineole/eucalyptol | 1034 | 1033 | 5.58 | 6.84 | 1.39 | 0.52 | |
| cis-β-ocimene | 1042 | 1040 | 0.36 | 0.53 | | | |
| γ-terpinene | 1061 | 1062 | 0.79 | | | | |
| cis-sabinene hydrate | 1069 | 1065 | 0.51 | 2.46 | | | |
| terpinen-4-ol | 1178 | 1177 | 2.04 | 0.51 | | | |
| α-terpineol | 1188 | 1189 | 0.49 | 0.33 | | | |
| dihydrocarveol | 1193 | 1193 | 8.60 | 7.07 | 0.80 | 1.17 | |
| dihydrocarvone | 1196 | 1194 | | 1.91 | | 0.30 | |
| trans-carveol | 1219 | 1217 | 0.37 | 0.43 | | 0.58 | |
| cis-carveol | 1229 | 1229 | 1.10 | 0.45 | | | |
| carvone | 1243 | 1242 | 53.06 | 26.44 | 0.71 | 0.90 | 0.43 |
| dihydrocarveol acetate | 1328 | 1326 | 3.72 | 8.03 | 12.96 | 7.80 | 0.58 |
| cis-carvyl acetate | 1367 | 1365 | 0.71 | 1.32 | 1.65 | 0.83 | |
| α-copaene | 1375 | 1376 | | | 0.42 | 0.83 | 1.25 |
| β-bourbonene | 1384 | 1384 | 1.00 | 3.92 | 18.12 | 27.43 | 35.57 |
| β-cubebene | 1390 | 1390 | | 0.37 | 0.24 | 0.32 | |
| β-elemene | 1391 | 1391 | | | 1.67 | 2.10 | 2.00 |
| β-caryophyllene | 1415 | 1418 | | 1.07 | 4.77 | 6.11 | 7.21 |
| β-gurjunene | 1426 | 1431 | | 0.40 | 1.83 | 2.70 | 3.15 |
| aromadendrene | 1442 | 1439 | | 0.39 | 1.98 | 3.08 | 4.29 |
| α-humulene | 1451 | 1454 | | | 0.39 | 0.49 | 0.49 |
| allo-aromadendrene | 1458 | 1458 | | 1.42 | 0.35 | 0.66 | 0.50 |
| cis-muurola-4-(14),-5-diene | 1462 | 1465 | 0.42 | 0.33 | 4.67 | 4.39 | 4.29 |
| γ-muurolene | 1477 | 1477 | | | 1.55 | 2.19 | 2.82 |
| germacrene-D | 1480 | 1480 | 1.43 | 4.34 | 13.41 | 9.75 | 10.33 |
| bicyclogermacrene | 1492 | 1494 | | 0.40 | 1.90 | 1.58 | 1.73 |
| γ-cadinene | 1511 | 1513 | | | 0.48 | 0.79 | 3.41 |
| cis-calamenene | 1522 | 1521 | | 0.49 | 2.03 | 3.52 | 4.03 |
| α-cadinene | 1539 | 1537 | | | 0.54 | 0.83 | 0.91 |
| viridiflorol | 1589 | 1590 | 0.49 | 1.32 | 4.01 | 3.80 | 2.61 |
| 1,10-di-epi-cubenol | 1614 | 1618 | | | 0.88 | 1.02 | 1.37 |

**Table 3.** *Cont.*

| Identified Compounds | RRI[a] | LRI[b] | Concentration (%)[c] | | | | |
|---|---|---|---|---|---|---|---|
| | | | (i) Spearmint | (ii) Soil Amended with Spearmint | | | |
| | | | | Days After Soil Amendment | | | |
| | | | | 15 | 30 | 60 | 90 |
| *t*-cadinol | 1644 | 1645 | | | | 0.45 | 0.70 |
| *α*-cadinol | 1657 | 1652 | | 0.30 | 0.65 | 1.04 | 0.50 |
| *α*-bisabolol | 1689 | 1685 | | | | 0.50 | 0.31 |
| mintsulfide | 1743 | 1744 | | | 0.30 | 0.69 | 0.76 |
| kaurene | 2029 | 2034 | | | | | 0.40 |
| Total identified compounds % | | | 95.59 | 96.53 | 94.56 | 94.01 | 89.64 |
| Spearmint Essential oil yield (%) | 3.26% | | | | | | |

[a] RRI, Relative Retention Index calculated relative to C7-C22 n-alcanes on HP-5 capillary column; [b] LRI, Literature Retention Index; [c] Components with percentage > 0.3% are presented.

**Table 4.** Qualitative and quantitative analysis of volatiles present (i) in oregano plant material, and (ii) in soil samples taken at different times after amendment with 4% (w/w) oregano tissue. Data averaged over two independent experiments.

| Identified Compounds | RRI[a] | LRI[b] | Concentration (%)[c] | | | | |
|---|---|---|---|---|---|---|---|
| | | | (i) Oregano | (ii) Soil Amended with Oregano | | | |
| | | | | Days after Soil Amendment | | | |
| | | | | 15 | 30 | 60 | 90 |
| a-thujene | 929 | 931 | 0.52 | 0.61 | 1.65 | 0.68 | |
| *a-pinene* | 936 | 937 | 0.85 | 0.65 | 1.75 | 1.19 | |
| *α*-camphene | 949 | 953 | | | 0.46 | 0.34 | |
| *β*-pinene | 976 | 980 | | 0.31 | 0.39 | | |
| *β*-myrcene | 990 | 991 | 1.76 | 0.67 | 0.34 | | |
| *α*-terpinene | 1015 | 1018 | 1.40 | 0.70 | 0.65 | 0.67 | |
| *p*-cymene | 1024 | 1026 | 6.27 | 5.56 | 15.11 | 13.54 | |
| limonene/*β*-phellendrene | 1029 | 1031 | 0.48 | 1.69 | 0.76 | 0.59 | |
| 1,8-cineole/eucalyptol | 1032 | 1033 | | 0.56 | | | |
| *γ*-terpinene | 1059 | 1062 | 4.26 | 0.86 | 0.30 | 0.55 | |
| cis-sabinene hydrate | 1066 | 1065 | | 0.55 | 1.78 | 2.01 | |
| trans-sabinene hydrate | 1094 | 1098 | | 0.30 | 0.99 | 1.31 | |
| borneol | 1163 | 1165 | 0.52 | 0.48 | 1.50 | 2.37 | |
| terpinen-4-ol | 1175 | 1177 | 0.89 | 1.05 | 1.75 | 2.57 | |
| dihydrocarveol | 1191 | 1193 | | 0.37 | 0.31 | | 0.98 |
| dihydrocarvone | 1195 | 1194 | | | | | 0.45 |
| trans-carveol | 1215 | 1217 | | | | | 0.60 |
| carvone | 1241 | 1242 | | 0.79 | | | 0.91 |
| thymoquinone | 1248 | 1250 | | 0.31 | 3.55 | | |
| thymol | 1293 | 1290 | 0.84 | 0.36 | 0.30 | 1.13 | |
| carvacrol | 1307 | 1298 | 78.31 | 78.52 | 58.86 | 48.22 | |
| dihydrocarveol acetate | 1331 | 1326 | | 0.46 | | | 0.90 |
| *α*-copaene | 1376 | 1376 | | | | | 0.93 |
| *β*-bourbonene | 1384 | 1384 | | | | | 38.15 |
| *β*-elemene | 1390 | 1391 | | | | | 2.45 |
| cis-caryophyllene | 1406 | 1408 | | | | | 0.46 |
| *a*-gurjunene | 1410 | 1409 | | | | | 0.49 |
| *β*-caryophyllene | 1416 | 1418 | 1.52 | 1.90 | 4.79 | 11.42 | 7.72 |
| *β*-gurjunene | 1425 | 1431 | | | | | 3.42 |
| aromadendrene | 1441 | 1439 | | | | | 4.62 |
| *α*-humulene | 1452 | 1454 | | | 0.55 | 1.27 | 0.52 |
| allo-aromadendrene | 1456 | 1458 | | | | | 0.79 |
| cis-muurola-4(14),5-diene | 1462 | 1465 | | | | | 4.97 |
| *β*-acoradiene | 1467 | 1467 | | | | | 3.06 |
| germacrene-D | 1476 | 1480 | | | | | 9.97 |
| bicyclogermacrene | 1492 | 1494 | | | | | 1.77 |
| *γ*-cadinene | 1513 | 1513 | 0.30 | 0.32 | 0.35 | 1.89 | 1.08 |
| cis-calamenene | 1520 | 1521 | | | | 0.39 | 3.61 |
| *α*-cadinene | 1538 | 1537 | | | | | 0.94 |
| caryophyllene oxide | 1582 | 1581 | | 0.31 | 0.53 | 2.27 | |
| viridiflorol | 1589 | 1590 | | | | | 2.14 |
| 1,10-di-epi-cubenol | 1612 | 1618 | | | | | 0.41 |

**Table 4.** *Cont.*

| Identified Compounds | RRI[a] | LRI[b] | Concentration (%)[c] | | | | |
|---|---|---|---|---|---|---|---|
| | | | (i) Oregano | (ii) Soil Amended with Oregano | | | |
| | | | | Days after Soil Amendment | | | |
| | | | | 15 | 30 | 60 | 90 |
| α-cadinol | 1656 | 1652 | | | | | 0.39 |
| mintsulfide | 1742 | 1744 | | | | 0.33 | 0.61 |
| Total identified compounds % | | | 97.92 | 97.33 | 96.67 | 92.74 | 92.34 |
| Oregano Essential oil yield (%) | | 6.90% | | | | | |

[a] RRI, relative retention index calculated relative to C7-C22 n-alcanes on HP-5 capillary column; [b] LRI, literature retention index; [c] Components with percentage > 0.3% are presented.

The amount of essential oil extracted from soil, was affected by the incorporated aromatic plant and the sampling time. Analysis of samples taken at 0 DAI showed that the essential oil yield was proportional to the incorporation rate of 4% (Table 2). Specifically, after the incorporation of 4% spearmint and oregano into the soil, the amounts of volatiles obtained at 0 DAI were 129.1 and 270.3 µL/100g, respectively. These yields are close to the expected ones (130 and 276 µL/100g), considering that the percentage essential oil yield of dried spearmint and oregano was 3.26 (Table 3) and 6.9 mL (Table 4), respectively. For both species, the total amount of volatiles present in the soil, decreased rapidly over time, with a reduction of 95% and 98% at 30 and 60 DAI, respectively (Table 2).

The GC/MS analysis, showed significant qualitative differences in the volatiles obtained from soil amended with spearmint or oregano, in samplings throughout different DAI (Tables 3 and 4). Specifically, the proportion of monoterpenes decreased significantly in time, whereas that of sesquiterpenes increased (although some of them are undetectable in the initial essential oil). Carvone, which is the main essential oil constituent, accounting for the 53% of the initial (original) spearmint essential oil, was decreased to 26% at 15 DAI and to 0.7% at 30 DAI (Table 3). Similarly, dihydrocarveol was decreased from 8.6% to 1.2% from 0 to 60 DAI, whereas at 90 DAI it was undetectable. In addition, the monoterpenes limonene and 1,8-cineole, reduced from 10.8% and 5.6% respectively at 0 DAI, to 3.9% and 0.5% at 60 DAI. One month later (90 DAI), limonene and 1,8-cineole were not detectable at all, in the soil. Almost all the monoterpenes were not detected in soil samples at 90 DAI.

The sesquiterpene group was determined sparingly in the essential oil of spearmint, represented by β-bourbonene, cis-muurola-4(14),-5-diene, germacrene-D and viridiflorol, which accounted for the 1, 0.4, 1.4, and 0.5% of the essential oil, respectively. Nevertheless, different sesquiterpenes were detected in significant amounts in soil mixed with spearmint plant material and in all sampling times (15, 30, 60 and 90 DAI) (Table 3). In most cases, their percentage was increasing in sampling time, especially up to 90 DAI. In particular, the amount of β–bourbonene was increased from 1% (0 DAI) to 35.6% (90 DAI), whereas this of germacrene-D was increase from 1.4% (0 DAI) to 10.3% (90 DAI). Additionally, β-caryophyllene, although undetectable in the essential oil of spearmint, it was accounted for the 1.1% and 7.2% of the soil volatiles at 15 DAI and 90 DAI, respectively. Considerable amounts of β-elemene, β-gurjunene, aromadendrene, γ-muurolene, γ-cadinene and cis-calamenene were also detected in soil mixed with spearmint, mostly at 60 and 90 DAI.

Conerning oregano, carvacrol was the main compound of essential oil (78.3%), followed by *p*-cymene (6.3%) and γ-terpinene (4.3%). (Table 4). The amount of carvacrol remained stable in the soil mixed with oregano at 15 DAI, though it was decreased at 30 and 60 DAI (58.9% and 48.2% respectively), and was not detected at all at 90 DAI. Additionally, *p*-cymene raised from 5.6% (15 DAI) to 15.1% (30 DAI) and decreased to 13.5% at 60 DAI, whereas γ-terpinene was decreased in the soil/plant mixture. The constitution of volatiles in the soil mixed with oregano at 15, 30 and 60 DAI revealed the monoterpene hydrocarbons α-camphene and β-pinene, which were not detected in oregano essential oil. On the other hand, most oxygenated monoterpenes, such as cis-sabinene hydrate, trans-sabinene hydrate, dihydrocarveol and thymoquinone were identified only in soil/oregano mixture (Table 4). In

general, the amounts of oxygenated monoterpenes was raised from 15 to 60 DAI. At 60 and 90 DAI, the volatiles from soil/oregano mixture were dominated by sesquiterpenes; *β*-caryophyllene, present in the original oregano oil at 1.5%, increased to 11.4% at 60 DAI, and decreased then to 7.7% at 90 DAI. It is furthermore noteworthy, that several sesquiterpenes not detected in oregano essential oil, were identified in great amounts in the volatiles obtained from soil/oregano mixture mostly at 90 DAI; the most of them were same with those obtained from soil/spearmint mixture. (Table 4). Particularly, the sesquiterpene *β*-bourbonene showed the most pronounced increase in the oil in relation to time, as though non-existent in the pure essential oil of oregano, at the last sampling held approximately 38.1%.

### 3.2. Protective Effect of Soil Amendment against Fusarium and Verticillium Wilt

Analysis of variance (ANOVA) applied on data obtained from average degree of infection (ADI) showed significant effects due to incorporated aromatic plant, species of fungi and their interactions with DAT (Table 5). Particularly, concerning the disease severity of inoculated plants with *Fol* or *Vd*, plants grown in soil amended with spearmint (MF and MV treatments) exhibited epinasty and chlorosis at 15 DAT, in contrast with inoculated controls (CF and CV) who had chlorosis and in some cases loss of leaf area <50% (Table 6). However, at 50 DAT, the same plants did not have macroscopic symptoms, although the respective controls infected with one of two pathogens exhibited chlorosis and loss of leaf area >50%. Contrarily, plants in ORF and ORV treatments developed differently over time: at 15 DAT, plants inoculated whatever the pathogen applied, exhibited epinasty and chlorosis, although at 50 DAT, ORF plants were unhealthy, presented chlorosis and loss of leaf area <50%, whereas ORV plants did not have macroscopic symptoms. The previous results strongly reflected at the AUDPC values: for both fungi, plants grown in soil amended with spearmint had up to 3.5 times lower AUDPC value in than that in the respective positive controls (Table 6). In the case of oregano, only the AUDPC for *Fol* inoculated plants was 2.6 times lower than that for the respective positive control.

**Table 5.** Results of analysis of variance applied on growth and physiological parameters, on productivity and on quality parameters of tomato, examined after the incorporation of spearmint and oregano into the soil and during the decomposition process. *F*-ratios' significance are given for the effects exerted by the experimental time (ET, the experiment being repeated twice), the incorporated aromatic plant (IAP), the species of fungi (SF) and the days after the transplantation (DAT).

| Variation Source | df | Shoot Length | Shoot Thickness | CCI[a] | A$_{net}$[a] | ADI[a] | df | Yield Per Plant | pH | °Brix[a] | Dry Matter |
|---|---|---|---|---|---|---|---|---|---|---|---|
| | | | | | | | | | | | Significance of *F*-Ratio |
| ET | 1 | NS | NS | NS | NS | NS | 1 | NS | NS | NS | NS |
| IAP | 2 | *** | *** | *** | *** | *** | 2 | *** | *** | *** | *** |
| SF | 2 | *** | *** | *** | *** | *** | 2 | *** | *** | *** | *** |
| IAP×SF | 4 | *** | *** | *** | *** | *** | 4 | *** | *** | *** | *** |
| ET×IAP | 2 | NS | NS | NS | NS | NS | 2 | NS | NS | NS | NS |
| ET×SF | 2 | NS | NS | NS | NS | NS | 2 | NS | NS | NS | NS |
| ET×IAP×SF | 4 | NS | NS | NS | NS | NS | 4 | NS | NS | NS | NS |
| DAT | 4 | *** | *** | *** | *** | NS | | | | | |
| DAT×ET | 4 | NS | NS | NS | NS | NS | | | | | |
| DAT×IAP | 8 | *** | *** | *** | *** | *** | | | | | |
| DAT×SF | 8 | *** | *** | *** | NS | *** | | | | | |
| DAT×ET×IAP | 8 | NS | NS | NS | NS | NS | | | | | |
| DAT×ET×SF | 8 | NS | NS | NS | NS | NS | | | | | |
| DAT×IAP×SF | 16 | *** | *** | *** | *** | *** | | | | | |
| DAT×ET×IAP×SF | 16 | NS | NS | NS | NS | NS | | | | | |

[a] CCI, chlorophyll content index; A$_{net}$, net photosynthetic rate; ADI, average degree of infection; °Brix, soluble solids content. *** significance at $p < 0.001$, NS = non-significant.

**Table 6.** Average degree of infection (ADI) (±se) and area under diseases progress curves (AUDPC) of transplanted tomatoes grown in soil amended with 4% (*w/w*) of spearmint or oregano plant material. Determination was made at 15, 30 and 50 days after inoculation which coinciding with the days after seedling transplantation (DAT). Full description of the treatments examined is found in Table 1.

| Treatment | Average Degree of Infection (ADI)[a] | | | AUDPC |
|---|---|---|---|---|
| | Days after Inoculation | | | |
| | 15 | 30 | 50 | |
| C | - | - | - | 0 |
| CF | 3.5 ± 0.10 a[b] | 4.9 ± 0.11 a | 4.8 ± 0.13 b | 160 |
| CV | 3.6 ± 0.13 a | 5.0 ± 0.12 a | 5.1 ± 0.09 a | 160.5 |
| M | - | - | - | 0 |
| MF | 2.2 ± 0.17 b | 1.3±0.09 cd | 1.2 ± 0.08 d | 51.25 |
| MV | 2.0 ± 0.19 b | 1.1±0.06 d | 1.1 ± 0.06 d | 45.25 |
| OR | - | - | - | 0 |
| ORF | 3.3 ± 0.16 a | 4.0 ± 0.13 b | 4.1 ± 0.16 c | 135.75 |
| ORV | 3.2 ± 0.13 a | 1.5 ± 0.10 c | 1.0 ± 0.04 d | 60.25 |

[a] Disease scored using a six-level disease index scale, where 1 = no disease symptoms, 2 = epinasty, 3 = chlorosis/losses of primary leaves, 4 = chlorosis / loss of leaf area <50%, 5 = chlorosis / loss of leaf area > 50%, 6 = practically dead plants. [b] Means in the same column followed by the same letter are not statistically significant different at *p < 0.05* according to Bonferroni adjusted LSD value (LSD$_B$ values for 15 days = 0.408, 30 days = 0.286, 50 days = 0.284).

### 3.3. Effect of Soil Amendment on the Development and Physiology of Tomato Plants

Analysis of variance (ANOVA) on data obtained from agronomical and physiological characteristics (tomato height, stem thickness, CCI and net photosynthetic rate) showed significant effects of the time of observation after seedling transplantation in the pots, incorporated aromatic plant, species of fungi and some of their interactions (Table 5). Similarly, ANOVA applied on data of qualitative traits (fruit yield/plant and qualitative parameters) showed significant effects due to incorporated aromatic plant, species of fungi and their interactions (Table 5).

Plants grown in soil amended with spearmint or oregano (M and OR treatments) had a significantly greater height at all determination times (12, 24, 36, 50 and 65 DAT), with the exception of the case of oregano at 12 and 24 DAT, where plants were up to 17% shorter than untreated controls (Figure 3). Among 36-65 DAT, plants in M and OR treatments were 30%–58% taller than control plants (C). Concerning the inoculated plants in all treatments (MF, ORF, MV, ORV), between 24 to 65 DAT, they were 34%–129% taller than the respective ones inoculated controls (CF and CV). The inoculated plants in all treatments (MF, ORF, MV, ORV), between 36 and 65 DAT, were 29%–51% taller than the healthy plants growing in untreated soil control (C).

Furthermore, spearmint and oregano soil incorporation (M and OR treatments) significantly increased stem thickness by 59% and 37%, respectively, compared to untreated control (Figure 4), with the exception of oregano-soil mixture at 12 DAT, where stem thickness was 24.5% smaller than the control. Generally, at inoculated plants, oregano and spearmint promoted the stem thickness with increasing DAT almost proportionally. In all treatments (MF, ORF, MV, ORV), between 24–65 DAT, the stem of inoculated plants was 19%–149% thicker than the respective ones at inoculated controls (CF and CV). Compared to the healthy controls (C), MF and MV treatments promoted stem thickness up to 52% and 45%, respectively, starting from 50 DAT. On the contrary, at all DAT, OF treatment caused a decrease of stem thickness up to 38%, compared to the untreated control (C). Similarly, OV treatment caused a decrease of stem thickness up to 37%, compared to the untreated control (C), until 24 DAT, but afterwards this is reversed.

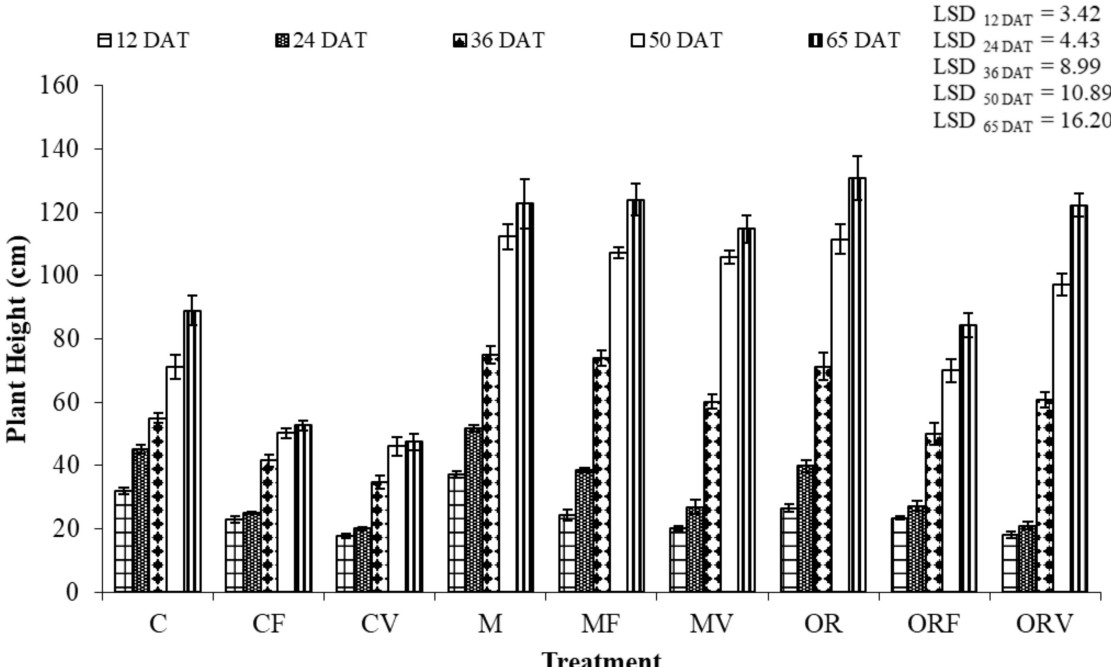

**Figure 3.** Plant height (means ± se) of healthy and inoculated transplanted tomatoes grown in soil amended with 4% (w/w) of spearmint or oregano plant material. Determination was made at 12, 24, 36, 50 and 65 days after the transplantation (DAT), whereas each mean represents the average of 24 determinations [12 replications (plants) × 2 experiments]. Full description of the treatments examined is found in Table 1.

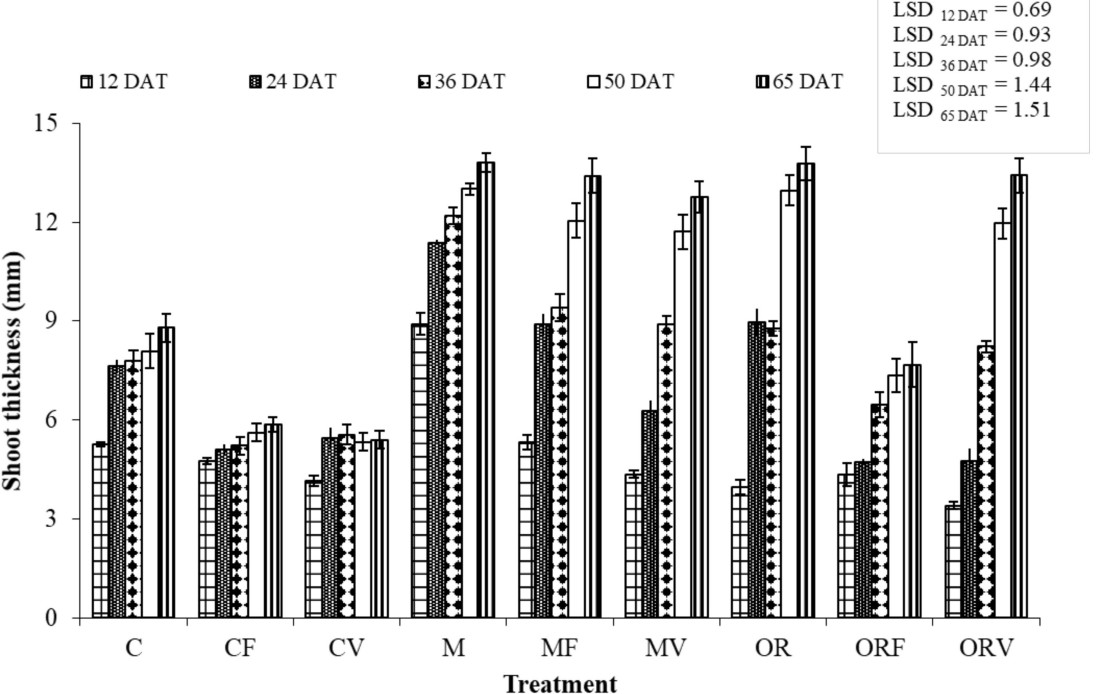

**Figure 4.** Shoot thickness (means ± se) of healthy and inoculated transplanted tomatoes grown in soil amended with 4% (w/w) of spearmint or oregano plant material. Determination was made at 12, 24, 36, 50 and 65 days after the transplantation (DAT), whereas each mean represents the average of 24 determinations [12 replications (plants) × 2 experiments]. Full description of the treatments examined is found in Table 1.

Chlorophyll content index (CCI) of tomato plants was significantly improved in the spearmint and oregano treatments (M and OR) ranging from 49.5% to 142% and 38% to 62%, respectively, at all determination times (12, 24, 36, 50 and 65 DAT), with the exception of plants grown in oregano-soil mixture, at 12 DAT, where CCI was 29% lower than the respective ones in the untreated controls (Figure 5). In the MF and ORF inoculated plants, CCI was 117%–761% and 108%–407%, respectively, greater than in the CF control plants. Similarly, in the MV and ORV inoculated plants, CCI was 162%–881% and 126%–890%, respectively, greater than in the CV control plants. Compared to the plants growing in untreated soil (C), MF and MV treatments promoted CCI up to 66% and 56%, respectively, starting from 36 DAT. On the contrary, at all DAT, plants in ORF treatment presented a decrease of CCI up to 26%, compared to the untreated control. However, in ORV treatment, CCI was increased by 16%–57% between 24–65 DAT, compared to the untreated control.

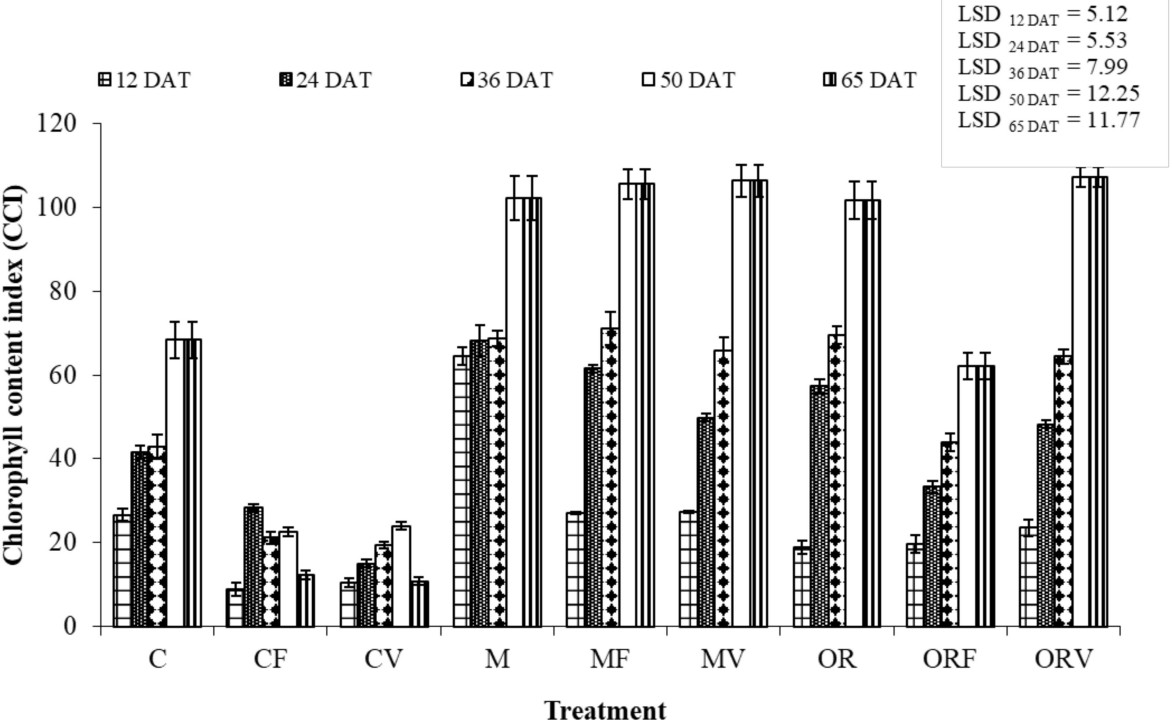

**Figure 5.** Chlorophyll content index (CCI) (means ± se) of healthy and inoculated transplanted tomatoes grown in soil amended with 4% (w/w) of spearmint or oregano plant material. Determination was made at 12, 24, 36, 50 and 65 days after the transplantation (DAT), whereas each mean represents the average of 24 determinations [12 replications (plants) × 2 experiments]. Full description of the treatments examined is found in Table 1.

The Photosynthetic rate ($A_{net}$) of tomatoes grown in M or OR treatment was significantly higher than in untreated controls at 30–45 DAT, ranged from 44%–81%, although at 15 DAT plants in OR treatment had 28.5% lower $A_{net}$ than the respective one at control (Figure 6). Regarding plants subjected to MF and ORF treatment, $A_{net}$ was 55%–980% and 126%–510%, respectively, greater than in the CF plants. Inoculated plants in MV or ORV treatment exhibited a comparable tendency to $A_{net}$, viz $A_{net}$ was 102%–968% and 57%–878%, respectively, greater than in the CV plants. Compared to the plants growing in untreated soil (C), $A_{net}$ was promoted in MF and MV treatments by 41%–80%, starting from 30 DAT, although previously (15 DAT) plants in the same treatments had up to 55% lower $A_{net}$ than the control. On the contrary, at all DAT, plants in ORF treatment presented 20%–63% inhibition of $A_{net}$ compared to the untreated control. Finally, ORV treatment promoted $A_{net}$ up to 79%, compared to the untreated control, but only after 45 DAT.

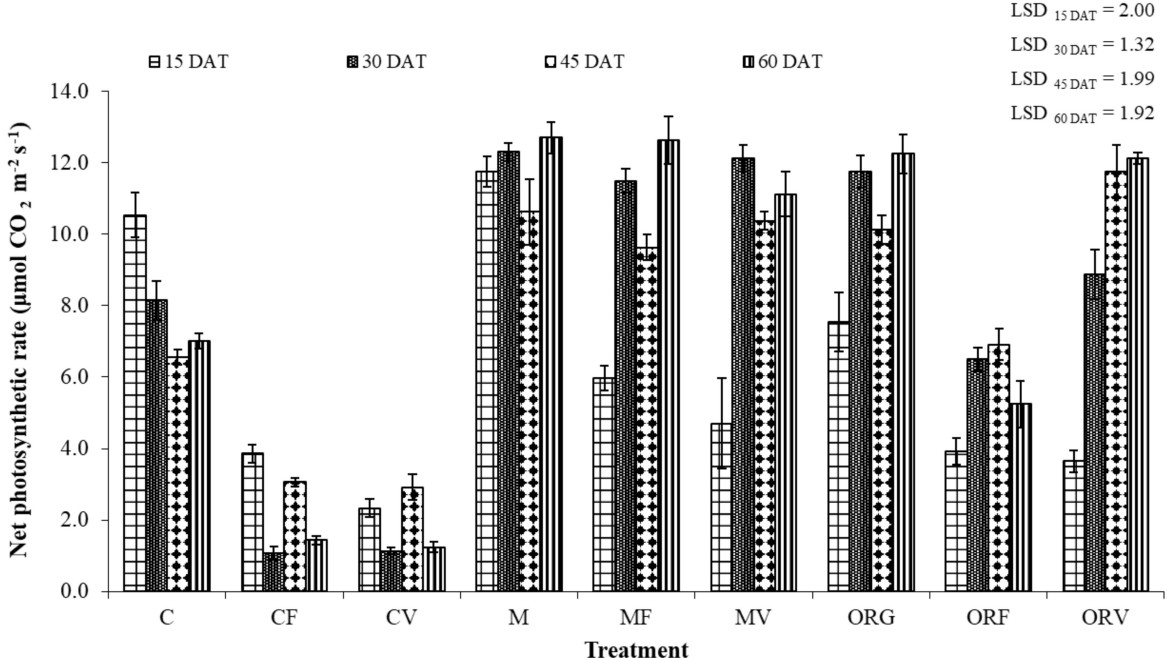

**Figure 6.** Net photosynthetic rate (means ± e) of healthy and inoculated transplanted tomatoes grown in soil amended with 4% (w/w) of spearmint or oregano plant material. Determination was made at 15, 30, 45 and 60 days after the transplantation (DAT), whereas each mean represents the average of 24 determinations [12 replications (plants) × 2 experiments]. Full description of the treatments examined is found in Table 1.

The yield per plant in M and OR treatments were increased by 103% and 116%, respectively, in comparison to the control plants. (Table 7). More impressively, the yield of plants in MF or MV treatment was increased by 1450% and 1130% compared with the respective infected controls, though compared with untreated healthy controls, MF or MV treatment increased yield by 90%–95%. A similar trend was observed in the yield of plants under ORV treatment, since it was increased by 1020% compared to infected controls and by 77% compared to the healthy controls. In contrast, the yield of plants in ORF treatment was increased by 360% compared with the infected controls, but decreased by 44% compared to healthy controls.

**Table 7.** Effect of soil amendment and seedling inoculation on fruit yield and quality. Means (±se) of yield per plant (kg) and physicochemical properties [pH, soluble solids content (°brix) and dry matter] of fruits harvested of plants grown in control soil (C) or in soil amended with 4% (w/w) of either spearmint (M) or oregano (OR) plant material. The seedlings were inoculated with either *Fol* (F) or *Vd* (V) or non-inoculated (no letter). Full description of the treatments is found in Table 1.

| Treatment | Yield Per Plant (kg) | Physicochemical Properties of Tomato Fruits | | |
| --- | --- | --- | --- | --- |
| | | pH | Soluble Solids Content (°brix) | Dry Matter (%) |
| C | 1130 [a] ± 107 c [b] | 3.92 ± 0.03 bc | 3.13 ± 0.04 c | 94.51 ± 0.12 bc |
| CF | 138 ± 31 e | 4.00 ± 0.02 abc | 2.30 ± 0.02 e | 94.79 ± 0.05 b |
| CV | 179 ± 38 e | 3.82 ± 0.03 c | 2.70 ± 0.05 de | 90.31 ± 0.07 e |
| M | 2290 ± 172 ab | 3.82 ± 0.02 c | 4.20 ± 0.05 b | 93.98 ± 0.12 cd |
| MF | 2144 ± 171 ab | 3.94 ± 0.02 bc | 4.40 ± 0.02 b | 93.39 ± 0.10 d |
| MV | 2202 ± 161 ab | 4.04 ± 0.00 abc | 5.00 ± 0.05 a | 94.50 ± 0.12 bc |
| OR | 2437 ± 167 a | 3.95 ± 0.05 abc | 3.20 ± 0.02 c | 94.61 ± 0.05 bc |
| ORF | 636 ± 85 d | 4.08 ± 0.02 ab | 3.10 ± 0.02 cd | 95.61 ± 0.05 a |
| ORV | 2004 ± 213 b | 4.17 ± 0.01 a | 4.10 ± 0.05 b | 94.31 ± 0.07 bc |

[a] Each mean represents the average of 24 determinations [12 replications (plants)×2 experiments]. [b] Means in columns followed by the same letter were not statistically significant different at *p < 0.05* according to Bonferroni adjusted LSD value (LSD$_B$ values for yield/plant = 312.2, pH = 0.223, °brix = 0.308, dry matter = 0.764).

The values of pH of fruits harvested from tomatoes grown in M and OR treatment did not differ from that in the untreated controls (Table 7). Similarly, pH of fruits harvested from plants in MF and ORF treatments did not differ from control plants infected with *Fol*, nor than that of healthy controls. The only differences were recorded on pH values were these of fruit harvested from plants in ORV treatment, were different from that of either untreated or treated with fungus controls (increase up to 0.35 units).

The soluble solids content (°brix) of fruits harvested from plants in M treatment was increased by 1.07 units compared with those in untreated controls (Table 7). Similarly, the soluble solids content of fruits harvested from plants in MF or MV treatment increased up to 2.30 units compared to the control plants infected with corresponding fungi, and up to 1.87 units compared to healthy controls. On the other hand, the soluble solids content of fruits harvested from plants in ORF or ORV treatment was increased by 0.8 and 1.4 units, respectively, compared to the control plants infected with corresponding fungi. Additionally, the soluble solids content of the fruits harvested from plants in ORV treatment was increased by 0.97 units compared with the healthy control plant.

The dry matter of fruit harvested from plants grown in M and OR treatments was not different than those in the untreated control plants. (Table 7). Similarly, the dry matter of fruits harvested from plants in MF or ORF treatment was slightly decreased by 1.4% and slightly increased by 0.82% compared to the respective infected controls. The relevant comparisons with untreated healthy controls, showed that dry matter of fruits harvested from plants in MF and ORF treatments was slightly decreased by 1.12% and slightly increased by 1.10%. The dry matter of fruits harvested from plants grown in ORV treatment was increased by 4.0% compared to the corresponding infected control, but did not differ from that in healthy control.

## 4. Discussion

### 4.1. Effect of Incorporation of Aromatic Plants at Soil Environment (C/N Ratio and Volatiles)

A good indicator for the maturity of composts is a C/N range between 15 and 20 [36]. In our work, C/N ratio was lower than 20 in the soil amended with spearmint or oregano at 60 DAI. Differences in C/N ratio values between soil amended with spearmint and oregano mainly at 0–30 DAI might be due to differences in C and N content of incorporated plant material, due to the decomposability of organic substrates [37], as well as due to the anatomical features of spearmint and oregano. The short stabilization of C/N ratio in soil amended with spearmint and oregano indicates that the direct incorporation of the plant material into the soil—without the preposterous time-consuming process of composting the aromatic plants—is possible. In tomato cultivation, the C/N values 38.5 and 27.2, recorded at 0 DAI, in soil amended with spearmint and oregano respectively, are much lower than those reported for wood-residue composts [38].

Essential oil composition of either spearmint or oregano changed dramatically with time. Oregano volatiles persist at high concentrations for a longer period than spearmint volatiles. This could possibly explain the inhibitory effect of oregano on tomato height (observed mainly up to 24 DAT), on stem thickness (at 12 DAT), on CCI (at 12 DAT) and on photosynthetic rate (at 15 DAT). The presence of monoterpenes in oregano oil–especially of oxygenated like carvacrol–has been previously related with plant growth inhibition [39]. Generally, monoterpenes decreased in time and almost all of them were not detected at 90 DAI and this is in agreement with results of Karamanoli et al. [40,41]. Based on the herein reported outcomes, further research is needed to shed light on whether the primary reason for the antifungal effects of spearmint and oregano into the soil is due to the presence of terpenes. The current results confirmed in vivo the findings of previous study published by Kadoglidou et al. [2], which carried out in vitro and in soil fungal cultures. In that study, *Vd* was the most sensitive fungi (both *Vd's* growth and sporulation were inhibited) among the four fungi tested, whereas fungicidal effects were exerted of the presence of either spearmint or oregano essential oils, as well as of the presence of their main components, carvone and carvacrol. In the current study, the low ADI values observed for *Vd*

(Table 6) may be partially attributed to the presence of carvone and carvacrol into the soil environment maintained until 15 and 60 DAI, respectively (Tables 3 and 4). Nevertheless, the high ADI value for *Fol* under oregano treatment in the current study is in contrast with the findings of Kadoglidou et al. [2], who concluded that oregano essential oil was very drastic against Fol. These differences between two studies could partially attributed to the type of the tested material of aromatic plant (dried biomass/essential oil/components of essential oil) and consequently their differences in corresponding rates. Concerning the mechanism of action of terpenes against pathogenic fungi (bacteria and yeasts also) this is not fully understood, but it is speculated to involve membrane disruption by the lipophilic compounds. Several studies have shown that monoterpenes exert membrane-damaging effects to microbial strains and also stimulates leakage of cellular potassium ions which provides evidence of a lethal action related to cytoplasmic membrane damage [42]. According to Trombetta et al. [43] monoterpenes perturb the lipid fraction of microorganism plasma membrane, resulting in alterations of membrane permeability and in leakage of intracellular materials, whereas Cox et al. [44] refers that they inactivate essential enzymes for physiological functions of microbes like respiration.

Regarding sesquiterpenes, they were detected in volatiles obtained from soil after spearmint and oregano incorporation, and in most cases at higher concentrations than in the initial oils. Monoterpenes and sesquiterpenes are found to act as allelochemicals which enhanced the plant defense, and several studies are focused on the isolation, introduction and overexpression of the responsible genes in genetically modified plants, to produce these terpenes [45]. Several researchers have related the occurrence of sesquiterpenes such as *β*-caryophyllene, *β*-elemene and *α*-humulene–which were detected in the current study—with root signalling on rhizosphere of cultivated plants [46,47], or as herbivore-induced plant volatiles, that attract natural enemies of the herbivores [48]. Sesquiterpenes, apart from their supporting role in plant defense, act as plant growth promoters in vegetables (representing D, E and F and 1,2-dehydrolactarolide A at lettuce seedlings) [49,50]. In the present study, it should be pointed out the long persistence of sesquiterpenes in the soil environment (mainly *β*-caryophyllene, *β*-bourbonene and germacrene-D), which could possibly explain the positive effects on tomato growth, on physiological characteristics, and on yield as well.

### 4.2. Effect of Incorporation of Aromatic Plants at Tomato Cultivation

There is a plethora of organic composted or uncomposted materials that can be used as soil amendments on tomato seedlings production or on transplanted tomatoes. In general, literature demonstrated the enhancement of tomato seedlings growth by using compost from forestry wastes [51], vine branches, grape prunings, husks and seeds [52], spearmint compost [17] and intact spearmint material [19]. In most cases, literature revealed that the stimulation effect on tomato seedlings growth is accompanied by an increase in soil fungal and bacterial population and an inhibition of weed emergence.

Concerning the effect of composts on transplanted tomato, literature generally demonstrated the enhancement of growth, yield and quality of tomato, as well as the improvement of soil properties. More precisely, these composts were vermicompost produced from cattle manure, market food waste and recycled paper waste [53], olive press cake, olive tree leaves and branches, vine branches and pressed grape skins [54], rice straw [55], vine branches, rice husks and straw and flax residues [56] and *Phragmites australis* plant material with animal manure [57]. Moreover, De Carvalho et al. [22] studied the allelopathic interactions in tomato intercropped with aromatic plants like basil, rue, fennel and mint and he found increases in yield per unit area. In current work, spearmint and oregano were incorporated into the soil without prior composting. Our findings indicate, that in contrast to high ratios of compost used in the most of the abovementioned studies, the spearmint and oregano material, incorporated into the soil at rate of 4% (w:w), enhanced significantly indices of growth and physiology of tomato as well as yield and soluble solids content of fruits. Specifically, compared to the untreated controls, the stem of tomato plants grown in soil amended with spearmint or oregano plant material was up to 58% taller and 59% thicker, whereas CCI was up to 142% higher, photosynthetic rate greater

up to 81%, the yield higher up to 116%, and the soluble solids content greater up to 1.07 units. It should be noted that at 12 and 24 DAT, the incorporation of oregano into the soil caused a significant but transient inhibition in most of the above parameters, probably due to the higher carvacrol concentration. These increases of values of the abovementioned parameters probably related to the high organic matter content in soil treated with spearmint and oregano. According to the literature, the enhancement of the abovementioned parameters, could be, at least partially, attributed to the higher soil microbial activity associated with the decomposition of the incorporated aromatic plants and the concomitant nutrient release [19,53,58]. In addition, increases in growth and yield could be attributed to plant growth regulators produced by microorganisms during the enhanced decomposition process [59]. Canellas et al. [60] and Scaglia et al. [61] referred to the occurrence of humates or hormones into vermicompost, since exchangeable auxin groups in humic acids were detected. An interesting point in the current study are the increases in total soluble solids in tomato fruits, which are rather related to the incorporation of spearmint into the soil than the disease pressure in both fungi (Table 7). These finding are in contrast to those reported of Papadaki et al. [32], who reported high total soluble solids in the fruits of "Early Pack" cultivar due to the high Vd pressure. Nevertheless, all these assumptions require further research and validation, especially in the case of decomposition of aromatic plants. The abundance of these aromatic plants—spearmint and oregano—in the Mediterranean ecosystem and their strong biological activity, combined with the abovementioned results in the current study, supports their potential use in soil management.

Organic soil amendments play important roles in the reduction of plant diseases caused by soilborne plant pathogens. With respect to the use of composts as soil amendment, it seems that composts may have highly suppressive effects against diseases caused by a variety of soilborne pathogens such as *Phytophthora* spp. [62], *Pythium* spp. [63], *Rhizoctonia* spp. [64] and *Fusarium* spp. [65]. Compost may suppress diseases in plants via a number of mechanisms, including antibiosis, competition, hyperparasitism and induction of systematic acquired resistance (SAR) in some host plants [66]. The antagonistic interactions with other fungi (like *Trichoderma* spp.) typically have been classified as based on antibiosis, mycoparasitism and competition for nutrients [67]. Moreover, the effectiveness of disease suppression is affected by parameters like moisture, pH and carbon to nitrogen ratio in compost [66].

In more detail, concerning *Fusarium* and *Verticillium* suppression into the soil using composts, Cotxarrera et al. [65] found that compost from vegetable and animal market wastes, sewage sludge and yard wastes showed a high ability to suppress *Fusarium* wild in tomato caused by *Fol* race 1, whereas Cheuk et al. [68] reported that *F. oxysporum* f.sp. *radicis-lycopersici* reduced when compost from yellow cedar sawdust used in greenhouse cultivated tomato, whose yield was improved by 74%. The co-compost of olive mill waste and leaves successfully suppressed Verticillium wilt in cotton and olive [69]. Giotis et al. [70] found that fresh Brassica tissue, household waste compost and composted cow manure significantly reduced soilborne disease severity (among them was *Verticillium albo-atrum*) and increased also tomato fruit yield. Additionally, Davis et al. [71] reported the suppression of *Verticillium* wilt of potato using maize as green manure crop, due to change in the activity and composition of the soil microflora. As far as we know, there are a few studies in the literature which investigate the effect of aromatic plants, as compost, green manure or co-culture plant, in order to reduce soilborne diseases. Among them, Chouliaras et al. [18] found that basil could be used as a co-culture plant, which can reduce the weed population and pathogenic soil organisms in organic farming, while exerting a positive effect on soil productivity. Kadoglidou et al. [19] and Chalkos et al. [17] reported that the incorporation of composted or uncomposted plant material of spearmint (intact dry material) has a positive effect in the soil environment (microbial density, action of nitrifying bacteria, soil respiration), whereas in vitro experiments [2] revealed that essential oils of spearmint and oregano as well as their main components carvone and carvacrol had inhibitory effect against *Vd* (fungicidal activity) and *Fusarium oxysporum* (fungistatic activity). Towards to the same direction, recently rosemary, clove and thyme oils were able to reduce *Fusarium* wilt in tomato [26]. In current

study, tomato grown in soil amended with 4% spearmint or oregano and inoculated with *Fol* or *Vd* had no macroscopic symptoms of disease after 50 DAT. Exception was the case of plants that were grown in soil amended with oregano and inoculated with *Fol*, as occurred symptoms of chlorosis and loss of leaf area up to 50%. The remaining inoculated plants were not only seemed healthy, but also showed significantly higher values of physiological indices (plant height and stem thickness) compared to healthy untreated controls (increases about 29%–53% after 50 DAT). The abovementioned tendency strongly reflected in measurements of CCI and photosynthetic rate of plants, wherein the inoculated plants showed significantly higher values in the above indices compared to healthy untreated controls (among 36–65 DAT increases of 50%–66% in CCI and 41%–80% in photosynthetic rate). Our findings show that plants were grown in soil amended with oregano and inoculated with *Vd* presented a time delay about 15 days to recovery, since, in comparison with the other treatments, they hold almost the lowest values concerning the evaluated parameters. Based on the above, we can assume that this phenomenon could be partially attributed to the long persistence of carvacrol into the soil environment (even after 60 DAI). Besides these impressive effects of spearmint and oregano incorporation into the soil, the yield of plants grown in soil amended with abovementioned aromatic plants and inoculated with *Fol* or *Vd* was increased by 77%–95% compared to healthy untreated controls, revealing that plants not only recovered from the initial inoculation but that they also benefited.

## 5. Conclusions

In the current study an innovative approach is proposed to exploit the aromatic plants towards a more sustainable and environmentally friendly tomato cultivation. According to our results, the direct incorporation of 4% spearmint or oregano plant material into the soil can improve tomato protection against soilborne fungi, plant growth, yield and product quality. Depending on the time of incorporation of aromatic plants and the climatic conditions, it requires a considerable lapse of time for the decomposition (two months in our case). Taken into consideration the cost of aromatic plants cultivation, their application in vast field crops would be rather difficult. However, since usually wilt diseases appear in defined spots in greenhouse or field area, initial material of spearmint or oregano could be transferred and incorporated from external inexpensive sources; i.e., plantation's residues, processing wastes etc. A valuable and smart solution would be the establishment of spearmint and oregano cultivation into rotation and cover crop system in organic farming. Moreover, further research is needed to elucidate the transformation of volatiles and the mode of action of secondary metabolites, liberated into the soil, after the incorporation of aromatic plants. The herein undertaken study demonstrates an innovative, potential use of aromatic plants, in the direction of a sustainable production system, which incorporates parts of the agricultural system in the overall ecosystem.

**Author Contributions:** Data curation, K.K.; formal analysis, K.K.; funding acquisition, K.K.; methodology, K.K., D.K., S.G., A.K. and P.C.; project administration, K.K.; supervision, K.K., D.K., P.C.; writing—original draft, K.K.; writing—review and editing, K.K., D.K., P.C. and E.M. All authors have read and agreed to the published version of the manuscript.

**Funding:** The project is included in the Act "Development projects of Research & Technology Innovation Development Projects (AgroETAK)" MIS 453350, under the OP "Development of Human Resources" (reconstruct, NSRF 2007-2013). The work is funded by the European Social Fund (ESF) and national resources (National Strategic Reference Framework, NSRF 2007–2013), coordinated by Institute of Plant Breeding and Genetic Resources, HAO–Demeter, Thessaloniki, Greece.

**Acknowledgments:** We express our thanks to Benaki Phytopathological Institute of Athens Greece, for providing *Fol* and *Vd* fungi strains.

**Conflicts of Interest:** The authors declare no conflict of interest.

**Abbreviations:** DAT: days after transplantation; DAI, days after incorporation of aromatic plants into the soil; *Fol*, *Fusarium oxysporum* f.sp. *lycopersici*; *Vd*, *Verticillium dahliae*; CCI, chlorophyll content index; A$_{net}$, Net Photosynthetic Rate; ADI, Average Degree of Infection.

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
