# Peer review of "Mentha and Oregano Soil Amendment Induces Enhancement of Tomato Tolerance against Soilborne Diseases, Yield and Quality"

_agronomy, doi:10.3390/agronomy10030406_

Round 1

Reviewer 1 Report

I attached comments.

Author Response

INSTITUTE OF PLANT BREEDING AND GENETIC RESOURCES

HELLENIC AGRICULTURAL ORGANIZATION-DEMETER (HAO-DEMETER)

PO Box 60458, 57001, THERMI THESSALONIKI

GREECE

Thessaloniki, 11th of March 2020

Dear Editor and Reviewers,

We appreciate the constructive and detailed remarks of the two reviewers of the manuscript entitled “Mentha and Oregano soil amendment induces enhancement of tomato tolerance against soilborne diseases, yield and quality” (Manuscript ID: agronomy-735469). We gladly faced the reviewers’ comments since this will undoubtedly improve our work. We appreciate their laborious evaluation. In addition to the revisions based on the reviewers’ comments, minor modifications where necessary.

Best Regards

Dr. Kadoglidou Kalliopi

Scientific Collaborator

Institute of Plant Breeding and Genetic Resources

Hellenic Agricultural Organization – “DEMETER”

Reviewer 2 Report

The manuscript entitled “Mentha and Oregano soil amendment induces enhancement of tomato tolerance against soilborne diseases, yield and quality” intends to evaluate the effects of two dried plant, mentha and oregano, in promotion of fungal resistance (Fusarium oxysporun f.sp. lycopersici (Fol) and Verticillium dalhiae (Vd)) of tomato plant. Overall, both dried plants into soil seem to be a natural way to use in alternative of synthetic fungicide into an organic farming system since the evaluated quality of tomato plants pointed out the positive results. Still, for manuscript quality improvement, the authors can find below a few observations.

Abstract

Line 18 – The author needs to correct the word “… concider among the most important horticulture crops…”

Introduction

Line 54 – The authors indicates the reference number 16, however at the end of paper, in References List (Line 690), a lack of information is evident such as the year, and number, volume of journal.

Line 61 – A similar situation regarding the description of reference number 23.

Line 78-79 – The terminology of fungal diseases: Fusarium oxysporun f. sp. lycopersici (Fol) and Verticillium dalhiae (Vd) did not showed the same identification during manuscript, e.g. in Line 27 –“…Fol or Vd…”, in Line 78 – “ V. dahlia and Fol…”. The authors need to follow the same terminology along the manuscript. Please confirm in all paper.

Materials and methods

Line 84 – The authors need to correct the sentences: “…and it was covered with with…”

Line 90 – The authors need to correct the sentences: “…, and stored fow two weeks…”

In section of production of plant material, the authors based their methodology in another study? If yes, the authors name needs to be added in this part.

Also, for better understanding the materials and methods applied I suggest a re-organization of the described points in this section of manuscript. For instance, in Line 107 the determination of organic carbon was referenced in the middle of soil amendment description. The authors after described the approach of the (2.2.) amendment soil and decomposition of plant, (2.3) production of tomato seedlings and transplanting to the amended soil, (2.4) production of inoculum of Verticillium / Fusarium and (2.5) Seedling inoculation and cultivation, should presenting the assessed methodologies regarding the C/N ratio determination as organic carbon and total nitrogen. Also, in Line 108 missing the reference of the macrokjeldahl methodology. The preparation and determination of mentha, oregano and soil samples for volatile constituents’ determination should be separated from the 2.2 point. I suggested the description of this methodology after the indication of determination of carbon and nitrogen determination.

Results and Discussion

Regarding the analysis of soil amended with aromatic plants, a comparison between both plants need to be evidenced since on soil amended with 4% of spearmint after 30 days shows a remarkable diminish of C/N compared to soil amended with 4% of oregano. Along the manuscript this difference needs to be evidenced in both sections results and discussion.

In all manuscript, the authors when indicated that determined value/results/parameters was significant, they need to add P < 0.05. For example in the Table presenting in the manuscript, for example in Line 235 “….in the soil, caused significant changes …”

Comparing the amounts of essential oil extracted from samples of soil amended with spearmint and oregano showed in Table 2, the authors need to correct the value obtained at day 0 in Line 245.

Another relevant indication it concerns the uniformity of the scientific results presentation, and for that the authors must follow the guidelines of the journal Foods.

The definition of letters that representing the significant difference in Table 6 must be corrected in Line 330 “…not statistically significant different at P < 0.05….” if is not statistically significant different is P > 0.05.

In Line 536 – 537 – the authors need to add the references that support the evidence regarding the effects of composts on transplanted tomato.

The legend of all figures, presented in the manuscript, is too extensive and have information that need to be removed to section of materials and methods. The authors must identify the graph presented and the rest of the information related to the experimental test must go to the materials and methods section so as not to become repetitive.

Author Response

(The authors gave the same response as above.)
